# Model for End-Stage Liver Disease Correlates with Disease Relapse and Death of Patients with Merkel Cell Carcinoma

**DOI:** 10.3390/cancers15123195

**Published:** 2023-06-15

**Authors:** Thilo Gambichler, Jürgen C. Becker, Laura Susok, Riina Käpynen, Nessr Abu Rached

**Affiliations:** 1Skin Cancer Center, Department of Dermatology, Ruhr-University Bochum, 44801 Bochum, Germany; laura.susok@kklbo.de (L.S.); riina.kaepynen@kklbo.de (R.K.);; 2Department of Dermatology and Phlebology, Christian Hospital Unna, 59423 Unna, Germany; 3Translational Skin Cancer Research, DKTK Partner Site Essen/Düsseldorf, West German Cancer Center, Dermatology, University Duisburg-Essen, 45122 Essen, Germany; j.becker@dkfz.heidelberg.de; 4German Cancer Research Center (DKFZ), 69120 Heidelberg, Germany; 5Department of Dermatology, Klinikum Dortmund gGmbH, 44137 Dortmund, Germany; 6International Centre for Hidradenitis Suppurativa/Acne Inversa (ICH), Department of Dermatology, Venereology and Allergology, Ruhr-University Bochum, 44801 Bochum, Germany

**Keywords:** MELD, MCC, carcinoma, liver, APRI-score, De Ritis, tumor metabolism, neuroendocrine, skin cancer, biomarkers

## Abstract

**Simple Summary:**

Merkel cell carcinoma (MCC) is a highly malignant skin tumor with high proliferation. Tumor metabolism is increasingly being studied. The liver is the central organ of metabolism. The aim of our retrospective study was to investigate liver scores (APRI, MELD, and De Ritis scores) for the clinical outcome of patients with Merkel cell carcinoma. We showed that the MELD score was a significant independent predictor of MCC relapse and MCC-specific. Calculation of the MELD score is useful for estimating clinical outcome and can easily be determined in daily clinical practice.

**Abstract:**

Merkel cell carcinoma (MCC) is a highly malignant skin tumor that occurs mainly in elderly and/or immunosuppressed patients. MCC prognosis has been significantly improved by the introduction of immune checkpoint inhibitor treatment. Recently, blood-based biomarkers have been investigated that can potentially predict the outcome of MCC patients. In this context, parameters of liver scores have not yet been investigated. We retrospectively recruited 47 MCC patients with available relevant laboratory data at primary diagnosis. At this time, we investigated blood-based scores as follows: model for end-stage liver disease (MELD), aspartate aminotransferase/platelet count ratio index (APRI), and the alanine transaminase/aspartate aminotransferase ratio (De Ritis ratio). MCC relapse was negatively correlated with the De Ritis score (r = −0.3, *p* = 0.024) and positively correlated with the MELD score (r = 0.3, *p* = 0.035). Moreover, MCC-specific death positively correlated with CCI score (r = 0.4, *p* = 0.01) and MELD score (r = 0.4, *p* = 0.003). In multivariable analysis, the MELD score remained in the regression model as significant independent predictor for MCC relapse (hazard ratio: 1.16 (95% CI 1.04 to 1.29; *p* = 0.008) and MCC-specific death (hazard ratio: 1.2 (95% CI 1.04 to 1.3; *p* = 0.009). We observed for the first time that the MELD score appears to independently predict both MCC relapse and MCC-specific death. These results should be further investigated in larger prospective studies.

## 1. Introduction

Merkel cell carcinoma (MCC) is a rare but highly malignant skin tumor that occurs mainly in elderly patients or patients with immunosuppression [1]. Fair skin type and high UV exposure are also risk factors for MCC. Merkel cell polyomavirus (MCPyV) is integrated in about 80% of MCC tumors [2]. A neuroendocrine and epithelial origin of MCC tumor cells is still being discussed [3]. MCC is characterized by very high recurrence rates within the first two years and frequent lymphogenic metastasis at initial diagnosis [4]. Moreover, the prognosis of MCC depends strongly on the tumor stage according to AJCC, localization of the metastases, number of metastases, occurrence of MCC recurrence, and the presence of a positive sentinel lymph node. The use of immune checkpoint inhibitors (ICIs) has significantly improved the MCC prognosis. However, about 50% of MCC patients do not respond to ICI therapy.

Several blood-based biomarkers for MCC outcome have already been investigated. For example, MCC relapse is associated with the level of pan-immune inflammation score (PIV) [5]. It has also been shown that higher neuron-specific enolase (NSE) levels during the course of the disease correlate significantly with the risk of MCC recurrence and death [6]. In recent years, tumor metabolism and the tumor environment have become more important in oncology research. It is believed that cancer cells can alter metabolism to maintain tumor nutrition acquisition [7]. MCC is a fast-growing skin tumor with a high proliferation rate, so the tumor also has high nutrient requirements [8]. The liver is the most important metabolic organ in the human body and it is responsible for the utilization, degradation, and excretion of metabolic products [9]. Altered tumor metabolic activity caused by MCC could also have an influence on liver metabolism. Even though blood-based prognostic biomarkers have recently been investigated in patients with MCC [5], parameters of liver metabolism have not been studied in this context. For example, the De Ritis quotient was shown to be an independent prognostic value for progression-free survival in neuroendocrine tumors [10]. We aimed to find out whether well-established baseline parameters of liver metabolism (APRI score, De Ritis score, and MELD score) are associated with disease outcome in patients with MCC.

## 2. Materials and Methods

We retrospectively recruited patients treated at the Skin Cancer Center of the Ruhr-University Bochum and having available laboratory results at primary diagnosis. The study was approved by the local ethics review board of the Medical Faculty of the Ruhr-University Bochum (#4749-13). The following blood-based scores were assessed at MCC diagnosis: model for end-stage liver disease (MELD) 3.78 × ln(serum bilirubin {mg/dL}) + 11.2 × ln(INR) + 9.57 × ln(serum creatinine {mg/dL}) + 6.43; aspartate aminotransferase/platelet count ratio index (APRI); alanine transaminase/aspartate aminotransferase ratio (De Ritis ratio). To classify the mortality risk of comorbidities, the Charlson Comorbidity Index (CCI) was calculated [11]. The CCI score was calculated based on age and comorbidities (history of myocardial infarction, chronic heart failure, hemiplegia, dementia, chronic kidney disease, cerebrovascular accident/TIA, connective tissue disease, peptic ulcer disease, COPD, liver disease, diabetes mellitus, leukemia, lymphoma, solid tumor, and AIDS). The patients were managed in accordance with the German guidelines for MCC [12]. Data analysis was performed using the statistical package MedCalc Software version 20.217 (MedCalc Software, Ostend, Belgium). Distribution of data was assessed by the D’Agostino–Pearson test. Non-normally distributed data was present as the median and range. Where appropriate, univariable analysis was performed using the Mann–Whitney test, Spearman correlation, Kendall’s Tau procedure, and Chi2 test. Multivariable analysis was carried out by Cox proportional hazard regression models including all significant parameters obtained from univariable statistics. We performed a ROC analysis to determine the cut-off values, area under the curve (AUC), and Youden index. *p*-values < 0.05 were considered significant. *p*-values were adjusted for multiple comparisons using the Benjamini–Hochberg method [13,14].

## 3. Results

### 3.1. Clinical Characteristics and Laboratory Values

In our retrospective study, we were able to include 47 patients with MCC. Clinical characteristics are detailed in Table 1. In our cohort, the median age was 78 years (range 51–95). The gender distribution of our patients was balanced (23 male and 24 female patients). About 42.6% of the patients had the primary tumor in the head and neck region, which is considered a high-risk area in MCC (n = 20). In the study, 18 patients were stage I (38.3%) and 14 were stage II (29.8%); 10 patients had a negative MCPyV status (21.3 %) and 37 had a positive MCPyV status (78.7%). About 30% of patients had an advanced stage at initial diagnosis (stage III 21.3%, n = 10 and stage IV 10.6%, n = 5). The liver value-based scores were: APRI score with a median of 0.3 (range 0.1–0.7), De Ritis score 1.2 (range 0.3–3), and MELD score 6.7 (range 5.3–20.1).

### 3.2. Clinical Outcome of Patients and Comorbidities

For clinical outcome, the events of MCC recurrence and MCC-specific death have been investigated (Table 2). About 45% of the patients (n = 21) showed MCC recurrence during the course of disease. The median time to MCC relapse was 11 months (range 2-122). MCC-specific death was present in 20 patients (38.3%). The median time to MCC-specific death was 30 months (range 3-122). To classify the mortality risk of comorbidities, the Charlson Comorbidity Index (CCI) was calculated (Table 2). Our MCC patients had a median CCI score of 7 (4–15). In early stages (stages I and II), the median CCI was 7 (range 4–9) and in the advanced stages (stages III–IV), the median CCI was 11 (range 7–15). The Mann–Whitney U test showed a significant difference in CCI score between early (I and II) and advanced stage (III and IV), so that patients with advanced stage possibly have a poorer prognosis due to comorbidities (*p* <0.001). The most common comorbidity used to calculate the CCI score was diabetes mellitus (n = 13; 27.6%), followed by a positive history of myocardial infarction (n = 10; 21.3%). In addition, 7 patients had dementia (14.9%), 5 patients had a cerebrovascular accident or transient ischemic attacks (10.6%), and 5 patients had chronic obstructive pulmonary disease (10.6%).

### 3.3. Univariable and Multivariable Statistics for MCC Outcome Measures

To determine the relationship of the different liver scores, we conducted univariable and multivariable regression models. On univariable analysis, MCC relapse was negatively correlated with the De Ritis score (r = −0.3, *p* = 0.02) and positively correlated with the MELD score (r = 0.3, *p* = 0.035). There was no correlation between CCI score and MCC relapse (*p* = 0.5). MCC-specific death was significantly associated with disease relapse (*p* = 0.003), disease stage at diagnosis (*p* = 0.018), and elevated C-reactive protein (*p* = 0.001). Moreover, MCC-specific death positively correlated with CCI score (r = 0.4, *p* = 0.01) and MELD score (r = 0.4, *p* = 0.003). After adjustment by the Benjamini–Hochberg method (Table 3), the *p*-values of the univariable tests (elevated CRP, MCC relapse, MELD score, CCI score, MCC stage at diagnosis) remained significant. Other clinical parameters, including age, gender, immunosuppression, APRI score, and Merkel cell polyomavirus status, were not significantly associated with MCC relapse or MCC-specific death (*p* > 0.05). In multivariable analysis (Table 4), the MELD score remained in the regression model as a significant independent predictor for MCC relapse (hazard ratio: 1.16 (95% CI 1.04 to 1.29; *p* = 0.008)) and MCC-specific death (hazard ratio: 1.2 (95% CI 1.04 to 1.3; *p* = 0.009)). Moreover, stage IV at diagnosis was another significant independent predictor for MCC-specific death (hazard ratio: 24.7 (95% CI 4.75 to 128.23; *p* = 0.0001)). At MCC diagnosis, 3 (6.4%) of 47 patients had liver metastases. However, MELD score did not significantly (*p* = 0.1) differ between patients with [median (range): 9.0 (7.5–19.7)] or without [median (range): 6.7 (5.3-20.1)] liver metastases.

### 3.4. Progression-Free Survival and MCC-Specific Death in Relation to MELD Score

We performed a ROC analysis to determine the cut-off values, area under the curve (AUC), and Youden index. The ROC analysis revealed a cut-off value of MELD score ≥ 10.94 for the event MCC relapse (*p* = 0.001, AUC specificity of 92.3%, sensitivity of 38.1%, and Youden index of 0.304). The progression-free survival of MCC patients is shown in Figure 1. In MCC, the disease most frequently recurs within the first two years. Progression-free survival (PFS) in the first two years was higher in the group with MELD score < 10.94 than in the group with ≥ 10.94 (PFS 59.4% vs. 17.1 %). For MCC-specific death, the ROC analysis showed a cut-off value ≥ 8.92 (*p* = 0.003) with an AUC of 0.7, specificity of 89.7%, sensitivity of 50%, and Youden index of 0.40. The survival probability in % of MCC patients with a MELD score < 8.92 and ≥ 8.92 is shown in Figure 2. Our analyses showed a 5-year survival probability with a MELD score < 8.92 of 69%. For a MELD score ≥ 8.92, the 5-year survival probability was only 20%.

## 4. Discussion

In recent years, there has been an increasing interest in tumor metabolism in the oncology field. It is suspected that cancer cells lead to altered metabolic activity and thus influence metabolic processes [7]. Increased metabolic activity and nutrient requirements caused by tumor cells also influence liver activity, which could lead to changes in liver parameters.

The De Ritis quotient is an established liver score that allows an assessment of the severity of liver cell damage. Some studies have found that the De Ritis quotient is also suitable as a prognostic marker in tumor disease [15,16]. In neuroendocrine tumors, the De Ritis quotient was related to PFS [10]. In addition, a correlation between PFS and the De Ritis quotient was found in bladder cancer [17]. The prognostic value and clinical utility of the De Ritis quotient in bladder cancer is controversial [17,18,19]. In prostate carcinoma, a higher De Ritis quotient was a predictor of worse pathological outcomes [20]. In COVID-19 patients, an increased De Ritis quotient was associated with increased mortality [21]. A pooled analysis with 9400 patients showed that the De Ritis quotient in solid tumors was significantly associated with poor clinical outcomes in relation to overall survival, cancer-specific survival, and recurrence-free survival [22]. A retrospective study could not show a correlation between the De Ritis quotient and skin tumors, but the skin tumor types have not been considered individually [23]. In our analysis, the De Ritis quotient correlated negatively with MCC relapse, but not with MCC-specific death. However, no significant correlation could be detected in the multivariable Cox regression analysis, so that the importance of the De Ritis quotient in MCC is considered low. The APRI score is a measure of the probability of developing liver fibrosis and cirrhosis [24,25]. The predictive value of the APRI score has been demonstrated in tumors only in hepatocellular carcinoma [26]. In our multivariable Cox regression analysis, no association was found between APRI score and relapse or MCC-specific death. Moreover, CCI score was not significant in the multivariable Cox regression, so that the comorbidities had no significant influence on the MCC prognoses.

MELD is widely used to predict short-term mortality among patients with cirrhosis who are on the waiting list for liver transplantation [27]. A high MELD score is a predictor of death, and therefore suggests an urgent need for liver transplantation [27]. It represents renal function, liver function, and part of the coagulation cascade in one parameter. The liver and kidneys are important for the detoxification and elimination of waste products. MCC is a fast-growing tumor that can have a high tumor metabolism and decay. The increased metabolism also causes more waste products to fall, which can interfere with coagulation, liver, and kidney function. The release of tumor-secreted factors also influences the function of various organ systems [28]. In addition, MCC causes altered metabolic activity, which could be reflected in MELD score. We believe that high tumor metabolism is reflected in a high MELD score. It is also possible that persons with pre-existing altered liver metabolism have a more fertile environment for MCC tumorigenesis and progression. However, this hypothesis needs further investigation. MELD appears to represent a simple score for an independent prognostication of MCC patients. We demonstrated that a high MELD score is independently associated with MCC relapse and MCC-specific death (independent of comorbidities, age, sex, viral status, tumor stage, MCC relapse). In the present study, however, tumor stage at initial diagnosis was significantly associated with MCC-specific death. Tumor stage IV at initial diagnosis was even associated with a high hazard ratio of 24.7 for MCC-specific death (95% CI 4.75 to 128.23; *p* = 0.0001). This link between prognosis and stage has also been described in the literature and was not unexpected [29]. By contrast, the effect size of MELD was very low as expressed by hazard ratios of about 1.2.

Nevertheless, the score could be useful in clinical practice for the detection of patients who have a worse prognosis and need closer monitoring. To our knowledge, this is the first examination of the MELD score in MCC patients. Possibly, the MELD score plays an important role in other tumor diseases as well. However, except for hepatocellular carcinoma, there are no reports on MELD and other tumor diseases. An important limitation of the present study is the retrospective design. We suggest that the MELD score should be investigated in a larger prospective study, including a larger sample size.

## 5. Conclusions

In conclusion, we observed for the first time that the MELD score can independently predict both MCC relapse and MCC-specific death. A MELD score > 10 could estimate MCC prognosis in clinical practice and indicates a worse prognosis. However, MELD score must also be prospectively validated on a larger sample size of MCC patients. For further research, the investigation of the metabolic activity of MCC cancer cells could be interesting.

## Figures and Tables

**Figure 1 cancers-15-03195-f001:**
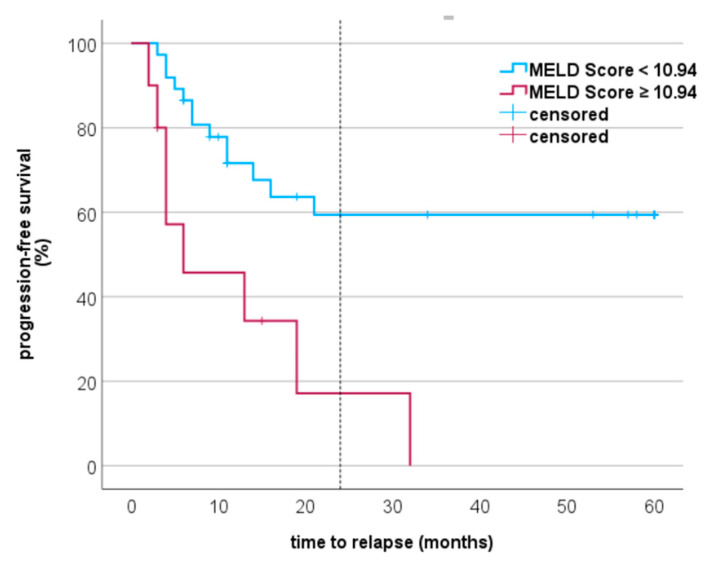
Kaplan–Meier curves of patients with Merkel cell carcinoma. The curves show that MELD score **≥** 10.94 was significantly associated with decreased progression-free survival in MCC (log-rank test: *p* = 0.001); The dashed line indicates month 24.

**Figure 2 cancers-15-03195-f002:**
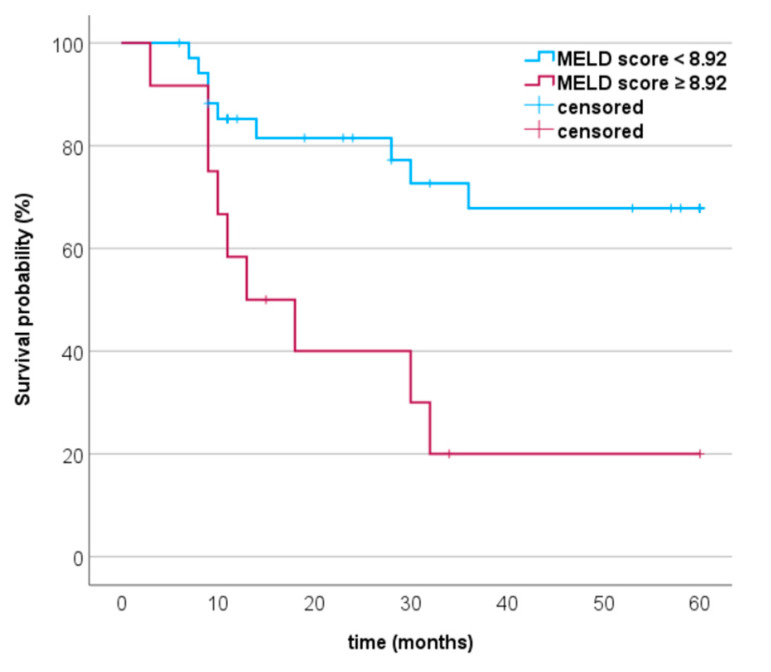
Kaplan–Meier curves of patients with Merkel cell carcinoma. The curves show that MELD score ≥ 8.92 was significantly associated with MCC-specific death (log-rank test: *p* = 0.003).

**Table 1 cancers-15-03195-t001:** Clinical characteristics at first diagnosis and laboratory values of patients with Merkel cell carcinoma (n = 47).

Parameters		Data
Age at diagnosis, median (range), years		78 (51–95)
Sex Male vs. female, n (%)		23 (49) vs. 24 (51)
Primary MCC Head/neck (no/yes), n (%) MCPyV (negative/positive), n (%) Lactate dehydrogenase (U/L), median (range) C-reactive protein Normal/elevated, n (%)		27/20 (57.4/42.6) 10 (21.3)/37 (78.7) 200 (109–699) 33 (70.2)/14 (29.8)
Tumor stage at diagnosis (according AJCC 2018), n (%)	Early stages Advanced stages	I 18 (38.3) II 14 (29.8) III 10 (21.3) IV 5 (10.6)
Parameters of liver metabolism, median (range)	APRI score De Ritis score MELD score	0.3 (0.1–0.7) 1.2 (0.3–3) 6.7 (5.3–20.1)

MCPyV, Merkel cell polyomavirus; APRI score, aspartate aminotransferase to platelet count ratio index; De Ritis score, alanine transaminase to aspartate aminotransferase ratio; MELD score, model for end-stage liver disease [3.78 × ln(serum bilirubin {mg/dL}) + 11.2 × ln(INR) + 9.57 × ln(serum creatinine {mg/dL}) + 6.43].

**Table 2 cancers-15-03195-t002:** Clinical outcome of patients and comorbidities of patients with MCC (n = 47).

Parameters		Data
MCC relapse MCC-specific	No MCC relapse, n (%) MCC relapse, n (%) Time to relapse, median (range), months No MCC-specific death, n (%) MCC-specific death, n (%) Time to death, median (range), months	26 (55.3) 21 (44.7) 11 (2-122) 29 (61.7) 20 (38.3) 30 (3-122)
CCI score, median (range)	All patients Stage I Stage II Stage III Stage IV	7 (4–15) 7 (4–9) 6.5 (5–9) 10.5 (7–14) 12 (7–15)
Comorbidities for CCI score, n (%)	History of myocardial infarction Congestive heart failure Peripheral vascular disease Cerebrovascular accident or TIA Hemiplegia Dementia COPD Connective tissue disease Peptic ulcer disease Moderate to severe liver disease Uncomplicated DM DM with end-organ damage Moderate to severe CKD Solid tumor (localized) Solid tumor (metastatic) Leukemia Lymphoma	10 (21.3) 3 (6.4) 3 (6.4) 5 (10.3) 1 (2.1) 7 (14.9) 5 (10.6) 4 (8.5) 1 (2.1) 1 (2.1) 8 (17) 5 (10.6) 1 (2.1) 32 (68.1) 15 (31.9) 2 (4.3) 2 (4.3)

CCI score, Charlson Comorbidity Index; TIA, transient ischemic attacks; COPD, chronic obstructive pulmonary disease; DM, diabetes mellitus; CKD, chronic kidney disease; AIDS, acquired immune deficiency syndrome.

**Table 3 cancers-15-03195-t003:** Adjustment of the univariable *p*-values using the Benjamini–Hochberg method.

Parameters	*p* Value	Rank	*p* Value (Adjusted)
Elevated CRP	0.001	1	0.01 *
MCC relapse	0.003	2	0.01 *
MELD score	0.003	3	0.01 *
CCI score	0.01	4	0.025 *
MCC stage at diagnosis	0.018	5	0.036 *
APRI score	0.15	6	0.25
Age	0.28	7	0.4
MCPyV	0.39	8	0.49
Immunosupression	0.78	9	0.87
Gender	0.91	10	0.91

CRP, C-reactive protein; MCC, Merkel cell carcinoma; CCI score, Charlson Comorbidity Index; APRI score, aspartate aminotransferase to platelet count ratio index; MELD score, model for end-stage liver disease; MCPyV, merkel cell polyomavirus; * significant result.

**Table 4 cancers-15-03195-t004:** Cox proportional hazards regression model for MCC-specific death (status: positive MCC-specific death; time: time to death) including variables from the univariable analyses with *p* value ≤ 0.05 (n = 47).

Parameters	Hazard Ratio (HR)	95% Confidence Interval (CI)	*p* Value
Elevated CRP	2.3	0.79–6.4	0.13
MCC stage at diagnosis	2.9	1.23–6.6	0.015 *
MCC relapse	2.1	0.64–6.7	0.22
CCI score	0.96	0.77–1.2	0.7
MELD score	1.2	1.04–1.3	0.009 *

CRP, C-reactive protein; MCC, Merkel cell carcinoma; CCI score, Charlson comorbidity index; APRI score, aspartate aminotransferase to platelet count ratio index; MELD score, model for end-stage liver disease; * significant result.

## Data Availability

Derived data supporting the findings of this study are available from the N.A.R. on reasonable request.

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
