# Peer review of "Model for End-Stage Liver Disease Correlates with Disease Relapse and Death of Patients with Merkel Cell Carcinoma"

_cancers, 2023, doi:10.3390/cancers15123195_

Round 1
Reviewer 1 Report
Gambichler and colleagues in their manuscript “Model for end-stage liver disease correlates with disease relapse and death of patients with Merkel cell carcinoma” retrospectively analyze a cohort of 47 MCC patients (21.8% virus-negative, 78.2% virus-positive) to determine if there are any correlations with disease progression or outcome with various markers of liver and kidney function. A previous study has found an association with the De-ritis score and neuroendocrine tumor PFS. This study identifies consistent modest but significant associations with MELD score, which is also prognostic for death in individuals needing liver transplantation. The study goes on to determine the predictive cutoff for PFS. This is an interesting, well written study to address whether altered metabolism has an effect on MCC outcome and is important for continued advancements in more personalized treatments of MCC. The study appropriately notes its limited size for the interpretation of its findings. Please see my specific comments for points that need to be addressed.
Major comments
1. Some of the results sections add up to 55 rather than 47. This should be corrected and all subsequent calculations should make sure they are using the correct sample set.
a. Virus-negative:12 + virus positive: 43 = 55
b. Stage I: 18 + Stage II: 14 + Stage III: 18* + Stage IV: 5 = 55
i. Stage III is only 10 in the table. The percentage is correct for 10 patients out of 47.
2. Due to the number of univariable tests performed, a false discovery rate correction should be implemented.
Minor comments
1. Page 2 line 66, “Altered metabolic activity caused by MCC could also have an influence on liver metabolism.” This sentence is a little unclear. Do you mean global altered metabolic activity, altered metabolic activity of the MCC, or something else? Please clarify.
2. Table 2. Exclude values that have no patients (i.e. AIDS and mild liver disease)
3. One of the many questions in carcinogenesis research is why some people develop cancer and others don’t even when they have similar lifestyles and risk exposure. Rather than MCC metabolism affecting liver metabolism, is it possible that individuals who have pre-existing altered liver metabolism have a more fertile environment for MCC tumorigenesis and progression? This is potentially likely since individuals with liver metastases did not show significantly differing MELD scores than those without. This counterpoint should be discussed.
Author Response
Gambichler and colleagues in their manuscript “Model for end-stage liver disease correlates with disease relapse and death of patients with Merkel cell carcinoma” retrospectively analyze a cohort of 47 MCC patients (21.8% virus-negative, 78.2% virus-positive) to determine if there are any correlations with disease progression or outcome with various markers of liver and kidney function. A previous study has found an association with the De-ritis score and neuroendocrine tumor PFS. This study identifies consistent modest but significant associations with MELD score, which is also prognostic for death in individuals needing liver transplantation. The study goes on to determine the predictive cutoff for PFS. This is an interesting, well written study to address whether altered metabolism has an effect on MCC outcome and is important for continued advancements in more personalized treatments of MCC. The study appropriately notes its limited size for the interpretation of its findings. Please see my specific comments for points that need to be addressed.
Answer: Thank you very much for your comments. We have revised the article.
Major comments
- Some of the results sections add up to 55 rather than 47. This should be corrected and all subsequent calculations should make sure they are using the correct sample set.
- Virus-negative:12 + virus positive: 43 = 55
- Stage I: 18 + Stage II: 14 + Stage III: 18* + Stage IV: 5 = 55
- Stage III is only 10 in the table. The percentage is correct for 10 patients out of 47.
Answer: We have improved the mistake.
- Due to the number of univariable tests performed, a false discovery rate correction should be implemented.
We adjusted the p-values according to the Benjamini-Hochberg method (see Table 3).
Minor comments
- Page 2 line 66, “Altered metabolic activity caused by MCC could also have an influence on liver metabolism.” This sentence is a little unclear. Do you mean global altered metabolic activity, altered metabolic activity of the MCC, or something else? Please clarify.
Answer: “tumor metabolic activity”
It is known that tumor diseases (especially tumours with high proliferation) show an increased tumor metabolism. Because the liver is a central organ for metabolism, the MCC could have an influence on global liver functions or laboratory parameters. We have changed this point.
- Table 2. Exclude values that have no patients (i.e. AIDS and mild liver disease)
Answer: We have changed this point.
- One of the many questions in carcinogenesis research is why some people develop cancer and others don’t even when they have similar lifestyles and risk exposure. Rather than MCC metabolism affecting liver metabolism, is it possible that individuals who have pre-existing altered liver metabolism have a more fertile environment for MCC tumorigenesis and progression? This is potentially likely since individuals with liver metastases did not show significantly differing MELD scores than those without. This counterpoint should be discussed.
Answer: We have added this important point to the discussion.
Your review comments could significantly improve the article. Thank you very much.
Reviewer 2 Report
The title of the manuscript describe the content of the article well, and the sample size is adequate.
The manuscript explain well the inclusion and exclusion criteria, the authors non report receiving ethics committee approval or informed consent from patients, if needed.
There are no ethical or regulatory issues and any conflicts of interest.
The data and method are adequately described. The result is adequately described, there is no redundancy between tables and text. In general, the manuscript is well done the objectives of the study are clear and results respond to these.
In the caption of table 3 the author declares Cox proportional-hazards regression model for MCC-specific death, but report OR and not HR (Hazard Rate).
Author Response
The title of the manuscript describe the content of the article well, and the sample size is adequate.
The manuscript explain well the inclusion and exclusion criteria, the authors non report receiving ethics committee approval or informed consent from patients, if needed.
There are no ethical or regulatory issues and any conflicts of interest.
The data and method are adequately described. The result is adequately described, there is no redundancy between tables and text. In general, the manuscript is well done the objectives of the study are clear and results respond to these.
Answer: Thank you very much for the review.
Information about the Ethics Committee and patient consent can be found on page 8.
Institutional Review Board Statement: The study was conducted according to the guidelines of the Declaration of Helsinki and approved by the local Ethics Committee of the Ruhr-University Bochum (#4749-13; 2013).
Informed Consent Statement: Informed consent was obtained from all subjects involved in the study. Written informed consent has been obtained from the patients to publish this paper
In the caption of table 3 the author declares Cox proportional-hazards regression model for MCC-specific death, but report OR and not HR (Hazard Rate).
Answer: Thank you for reading carefully. We have corrected this.